# Antimicrobial Activity of Selenium Nanoparticles (SeNPs) against Potentially Pathogenic Oral Microorganisms: A Scoping Review

**DOI:** 10.3390/pharmaceutics15092253

**Published:** 2023-08-31

**Authors:** Eulàlia Sans-Serramitjana, Macarena Obreque, Fernanda Muñoz, Carlos Zaror, María de La Luz Mora, Miguel Viñas, Pablo Betancourt

**Affiliations:** 1Center of Plant, Soil Interaction and Natural Resources Biotechnology, Scientific and Biotechnological Bioresource Nucleus (BIOREN-UFRO), Universidad de La Frontera, Temuco 4811230, Chile; mariluz.mora@ufrontera.cl; 2Center for Research in Dental Sciences (CICO), Endodontic Laboratory, Faculty of Dentistry, Universidad de La Frontera, Temuco 4811230, Chile; m.obreque06@ufromail.cl (M.O.); f.munoz24@ufromail.cl (F.M.); 3Department of Pediatric Dentistry and Orthodontics, Faculty of Dentistry, Universidad de La Frontera, Manuel Montt #112, Temuco 4811230, Chile; carlos.zaror@ufrontera.cl; 4Center for Research in Epidemiology, Economics and Oral Public Health (CIEESPO), Faculty of Dentistry, Universidad de La Frontera, Temuco 4811230, Chile; 5Laboratory of Molecular Microbiology & Antimicrobials, Department of Pathology & Experimental Therapeutics, Faculty of Medicine & Health Sciences, University of Barcelona, 08907 Barcelona, Spain; mvinyas@ub.edu; 6Department of Integral Adultos, Faculty of Dentistry, Universidad de La Frontera, Temuco 4811230, Chile

**Keywords:** selenium nanoparticles, antimicrobial activity, biofilm, planktonic state, oral pathogens

## Abstract

Biofilms are responsible for the most prevalent oral infections such as caries, periodontal disease, and pulp and periapical lesions, which affect the quality of life of people. Antibiotics have been widely used to treat these conditions as therapeutic and prophylactic compounds. However, due to the emergence of microbial resistance to antibiotics, there is an urgent need to develop and evaluate new antimicrobial agents. This scoping review offers an extensive and detailed synthesis of the potential role of selenium nanoparticles (SeNPs) in combating oral pathogens responsible for causing infectious diseases. A systematic search was conducted up until May 2022, encompassing the MEDLINE, Embase, Scopus, and Lilacs databases. We included studies focused on evaluating the antimicrobial efficacy of SeNPs on planktonic and biofilm forms and their side effects in in vitro studies. The selection process and data extraction were carried out by two researchers independently. A qualitative synthesis of the results was performed. A total of twenty-two articles were considered eligible for this scoping review. Most of the studies reported relevant antimicrobial efficacy against *C. albicans*, *S. mutans*, *E. faecalis*, and *P. gingivalis*, as well as effective antioxidant activity and limited toxicity. Further research is mandatory to critically assess the effectiveness of this alternative treatment in ex vivo and in vivo settings, with detailed information about SeNPs concentrations employed, their physicochemical properties, and the experimental conditions to provide enough evidence to address the construction and development of well-designed and safe protocols.

## 1. Introduction

Globally, the most prevalent bacterial diseases that impact the human oral cavity are dental caries, periodontal disease, and pulp and periapical infections [1]. A wide variety of bacteria are involved in such oral pathologies, including *Streptococcus mutans*, *Porphyromonas gingivalis*, and *Enterococcus faecalis* [2,3,4], whereas the most frequent oral fungal infection encountered in general dental practice is candidiasis, which is mainly caused by *Candida albicans* [5]. Many oral pathogenic microorganisms evolve from their planktonic state and band together as “matrix-enclosed communities” to form self-formed mucilaginous structures known as biofilms [6,7]. The structural and physiological features of microbial biofilms significantly contribute to resistance to antimicrobial agents and the host’s immune response [8,9], which can lead to the development of chronic infections and contribute to treatment failure [10]. Indeed, the increase in virulence and antibiotic resistance of microorganisms living in oral biofilms is correlated with the formation of an extracellular matrix (a physical barrier) [11] and the acquisition of antibiotic resistance genes (a biological barrier) [12], which leads to an increase in the difficulties in treatment and economic costs [13,14]. In this scenario, various approaches have been proposed for researchers, clinicians, and companies to overcome these difficulties, including the urgent need to discover or generate novel antimicrobials that can outpace microbial resistance mechanisms. In this regard, nanotechnology has emerged in recent years as a novel strategy to combat pathogenic microorganisms, offering a viable alternative to conventional antimicrobials with very good prospects [15,16,17]. Nanoparticles (NPs) can either be used as direct bactericidal agents [18,19] or as carriers of other therapeutic agents around or into bacterial cells [20,21,22]. As such, metal and metal oxide-based NPs (e.g., silver, gold, copper, zinc, and all their oxide derivatives) have been the most frequently investigated nanomaterials displaying inherent antimicrobial properties. It should be mentioned that some studies have reported that various metallic NPs exhibit toxicity [23] and promote the spread of antimicrobial resistance [24,25,26]. Accordingly, metalloid-based NPs (e.g., selenium and tellurium) have attracted increasing interest due to their intermediate nature between metals and non-metals, enabling their use in various applications [27]. Metalloids can exert several effects on cells and tissues and are considered valuable tools for diagnosing, treating, and preventing diseases [28,29].

This work will focus mainly on metalloid NPs made of selenium (Se): selenium nanoparticles (SeNPs). SeNPs have gained worldwide attention for their high degree of absorption, high biological activity, low toxicity, and considerable efficiency in hindering oxidative damage compared to their Se-based counterparts [30,31]. In the biomedical field, SeNPs potentially offer a beneficial role due to their antioxidant [32,33] and anticancer properties [34,35], antimicrobial activity [36], and immunoregulatory properties [33]. It has been widely reported that SeNPs have a broad spectrum of activity against bacteria and fungi [37,38]. Their antimicrobial capability may be associated with the overproduction of reactive oxygen species (ROS), which leads to cell membrane damage, the inhibition of amino acid synthesis, and the blockage of DNA replication [39]. These relevant qualities have prompted researchers to evaluate the use of SeNPs as a promising tool to combat multidrug-resistant bacteria and other microbial pathogens. Nevertheless, the published studies examining the antimicrobial action of SeNPs exhibit significant heterogeneity and considerable variability in their methodological approaches and results. This variability is primarily attributed to differences in the synthesis methods employed, the size of the SeNPs, the concentration of the SeNPs tested, the bacterial mode of life (planktonic or sessile), and the specific microbial species investigated.

Thus, considering all the above, this scoping review aimed to provide a detailed look into the potential of SeNPs as tools for combating microbial pathogens causing oral infectious diseases.

## 2. Materials and Methods

### 2.1. Protocol and Registration

This scoping review was conducted following the guidelines outlined in the Preferred Reporting Items for Systematic Reviews and Meta-Analyses Extension for Scoping Reviews (PRISMA-ScR) [40]. The review protocol is accessible and can be obtained at https://osf.io/4ebqf/?view_only=60b10cbdf57d4d73925f523f401e9b33 (accessed on 25 November 2022).

### 2.2. Eligibility Criteria

We included primary ex vivo, in vivo, and in vitro studies, which were published in English, Spanish, or Portuguese, to assess the association between the use of SeNPs and antimicrobial activity against the most prevalent microorganisms associated with oral pathogens (*Streptococcus mutans*, *Actinomyces* spp., *Candida albicans*, *Porphyromonas gingivalis*, *Prevotella* spp., *Streptococcus oralis*, *Aggregatibacter* spp., *Enterococcus faecalis*, and *Fusobacterium nucleatum*). No limitations were imposed on the publication dates or study locations. Narrative and systematic reviews, clinical studies, letters to the editor, opinion pieces, and conference abstracts were excluded.

### 2.3. Sources of Information and Search Strategy

A systematic literature search was conducted up to 5 May 2022 using the MEDLINE, Embase, Scopus, and Lilacs databases. The search strategy utilized for each database can be found in the Appendix A. We examined the reference lists of the included articles and previous systematic reviews to identify other possible studies that could be included.

### 2.4. Selection of Sources of Evidence

The entire list of identified references was imported into an EndNote X9 database, simplifying their organization and enabling the removal of duplicate entries. The selection process and data extraction were carried out using the Rayyan online software (http://rayyan.qcri.org) accessed on 5 May 2022. The blind mode was enabled to ensure that the activities of each reviewer remained concealed from the others. Two reviewers (F.M. and M.O.) individually performed the selection of studies based on their titles and abstracts. Subsequently, they independently assessed the full texts of the identified studies, adhering to the pre-defined eligibility criteria. Discrepancies were resolved by consensus. The reviewers were not kept unaware of the authors or journals of the identified studies and the reasons for excluding certain studies were carefully documented.

### 2.5. Data Charting Process

Two reviewers (F.M. and M.O.) conducted the data extraction process. They used a pre-defined Excel form to extract the following information from each article: study identification details, study type, study objective, sample size, study model, description of the study model, incubation time, microorganism employed, organizational form of the studied bacteria, type of SeNPs, concentration of SeNPs, size of SeNPs, measures of effectiveness and safety, comparator, side effects, results, and principal conclusions of the study. In case of any disagreements during the extraction, the reviewers resolved them through discussion and reached a mutual agreement.

### 2.6. Critical Appraisal

The chosen articles were independently analyzed by three reviewers (E.S.-S., F.M., and M.O.) utilizing the Toxicological Data Reliability Assessment Tool (ToxRTool) [41]. The analysis was performed using a predefined Microsoft Excel^®^ (version 16.19) file, which includes two distinct sections—one dedicated to in vivo data and the other to in vitro data. The tool used for the evaluation of in vitro studies comprises 18 criteria. In addition, the criteria used for the assessment were divided into five distinct groups: I. test substance identification; II. test system characterization; III. study design description; IV. study results documentation; and V. plausibility of study design and data. Each criterion can be assigned either a “1” (one point), i.e., “criterion met”, or a “0” (no points), i.e., “criterion not met”. Based on the total points assigned to each study, a reliability category (ranging from 1 to 3) was proposed. In vitro studies awarded 15–18 points were placed in the first category, 11–14 points in the second category, and less than 11 in the third category. Moreover, a fourth category labeled “not assignable” was included to account for cases where the documentation provided in the studies was insufficient or derived from secondary sources such as reviews, handbooks, or other non-primary research sources. Furthermore, the tool emphasized the minimum information requirements considered essential for a study to be deemed reliable by highlighting them in red. In detail, as per the tool’s guidelines, a study was assigned a data reliability category of 1 or 2 only if it met the specific criteria rated as “1”, regardless of the overall total score obtained [41].

### 2.7. Synthesis of Results

The results were synthesized based on the guidelines provided by Green et al. [42]. Indeed, a narrative overview model was developed to provide a comprehensive synthesis of the previously published studies. Tables were utilized to present comprehensive information on the antimicrobial effectiveness of SeNPs against planktonic and biofilm microorganisms, as well as any potential side effects observed in in vitro studies.

## 3. Results

### 3.1. Selection of Sources of Evidence

A sum of 1019 articles were found in electronic database searches. After removing 328 duplicates, 691 articles remained. Among them, 668 were excluded based on the assessment of their title and abstract, leaving 23 articles for full-text evaluation. Four of these articles were excluded due to unavailability of their full texts. Through a manual search, eight additional articles were included. Finally, a total of 27 articles were deemed eligible for this scoping review. Figure 1 illustrates the selection process in a flow diagram.

#### 3.1.1. Characteristics of Sources of Evidence

All of the articles included in this review had an in vitro design, and they were published between the years 2015 and 2023. Among the selected studies, eleven were from India, five from Iran, five from Egypt, two from the United States, and one each from Italy, Brazil, Japan, and Serbia. The distribution based on the microbial species associated with oral infections that were tested and their growth mode were as follows: twenty articles evaluated the antimicrobial effect of SeNPs on the fungus *C. albicans*, eight articles on *E. faecalis*, seven articles on *S. mutans*, and one article on *P. gingivalis.* Nineteen studies performed assays in planktonic microorganisms, two used biofilms, and six incorporated both planktonic and biofilm cultures. Therefore, there were 25 articles assessing the bioactivity of SeNPs in planktonic growth mode and 8 articles in biofilm forms, with the *C. albicans* being the most studied microorganism. Eighteen articles utilized SeNPs synthesized by biological approaches, whereas ten studies employed SeNPs produced by synthetic approaches, including physical and chemical techniques. Most of the studies included reported using SeNPs of varying particle sizes; only two articles did not mention the particle size [43,44]. Eighteen articles reported using SeNPs with an average size smaller than 100 nm, whereas six articles tested the efficacy of SeNPs with diameters larger than 100 nm, with all of them possessing a spherical morphology. Most of the studies reported the concentrations of SeNPs employed to test their antimicrobial efficacy, except six. Thirteen articles used SeNPs concentrations ranging from 5 to 500 μg/mL, whereas eight studies used even higher SeNPs concentrations, up to 5000 μg/mL (Appendix A).

#### 3.1.2. Critical Appraisal of Sources of Evidence

In the overall assessment, nineteen articles were regarded as reliable without restrictions, two were categorized as reliable with restrictions, and six were identified as not reliable. Appendix A presents the methodological quality assessment of each article. In the “test substance identification” criterion group, one article did not report the purity of the SeNPs, three articles presented no information about the physicochemical properties, and four articles provided no information on the source of the SeNPs. This lack of information seriously compromises the transparency of the experimental work and can affect the quality of the results. In the second criterion group, “test system characterization”, eight articles did not report adequate information on the source of the testing system. Concerning the “study design description”, six articles failed to provide the essential minimum information required in the “description of study design”, resulting in their classification as not reliable. Four studies did not mention the SeNP concentrations in the applied media. Three articles did not describe the negative controls employed in the experimental procedure, and three did not include positive controls. It is worth noting that appropriate and accurate controls are crucial when investigating the bioactivity of antimicrobial agents using susceptibility determinations. Otherwise, there is no confidence that the experiments worked. Hence, it is crucial to have a comprehensive description to accurately interpret the observed effects. Within the fourth criterion group, “study results documentation”, eleven articles did not report the statistical methods used for data analysis, three articles lacked a complete and transparent description of the study results, and one did not provide a clear description of the study endpoint(s) and its method of determination. In the fifth and final criterion, “plausibility of study design and data”, three articles did not achieve the maximum score due to issues related to selective outcome reporting.

### 3.2. Synthesis of Results

#### 3.2.1. Antimicrobial Efficacy on Planktonic Microorganisms

In twenty-five studies, the antimicrobial effectiveness on planktonic microorganisms was assessed using various parameters such as the size of the inhibition zone, the minimum inhibitory concentration (MIC), the minimum bactericidal concentration (MBC), and number of colony-forming units (CFU/mL). The key findings of these evaluations are summarized in Table 1. In eleven studies, the inhibition zone increased in a concentration-dependent manner. Interestingly, certain authors reported that the antimicrobial efficacy of SeNPs in inhibiting microbial growth was either equivalent to or even higher than the one displayed by conventional antibiotics, in addition to exerting a synergistic effect when combined with antibiotics (e.g., nystatin) or other natural compounds (e.g., plant extracts). In general, the antimicrobial activity of SeNPs against *C. albicans* was stronger than the one observed against *E. faecalis* and *S. mutans*, with lower MIC values in most cases. Regarding the minimum bactericidal concentration (MBC), three studies showed a bactericidal effect of SeNPs on *E. faecalis*, *S. mutans*, and *C. albicans*. In one study, the MBC of SeNPs for *E. faecalis* was higher than the control (ampicillin). In one study, the results showed that the SeNP concentrations were not sufficient to kill *P. gingivalis* bacteria. In two studies, according to MBC values, the exposure time increased the bactericidal capacity against *S. mutans*, *C. albicans*, and *E. faecalis*. In one study, the number of viable cell counts was reduced compared to the control through the application of SeNPs in combination with antimicrobial photodynamic therapy (aPDT) against *E. faecalis* with a concentration of 128 μg/mL; however, for *S. mutans*, this reduction was not observed. One study also showed a reduction in *P. gingivalis* viable cell counts with increasing concentration. Most studies showed that SeNPs had antimicrobial activity against planktonic cultures of *C. albicans*, *S. mutans*, *E. faecalis*, and *P. gingivalis.*

#### 3.2.2. Antimicrobial Efficacy on Biofilm Microorganisms

The antimicrobial efficacy on microbial biofilms was evaluated in eight studies through the percentage of biofilm inhibition, percentage of viable cells, and number of colony-forming units (CFU/mL). The main results are shown in Table 2. In three studies, the percentage of biofilm inhibition demonstrated an incremental rise with increasing concentrations of SeNPs. The highest inhibition was observed in the case of *C. albicans*, reaching a 99% inhibition rate. However, one article surprisingly reported that the biogenic SeNPs tested did not provide a considerable antibiofilm effect against *C. albicans*. In this regard, the authors mitigated those drawbacks by preparing SeNP@PVP_Nystatin nanoconjugates with improved antibiofilm activities. Moreover, in one study, the number of viable cell counts decreased significantly when SeNPs were used in combination with photodynamic therapy against *E. faecalis*. Specifically, SeNP concentrations of 64 and 128 μg/mL of SeNPs showed substantial reductions in viable cell counts compared to the control group; however, for *S. mutans*, this reduction was not observed. In one study, the percentage of viable cells at 24 and 48 h was lowest in SeNPs compared to the control. Overall, the majority of the studies demonstrated that SeNPs exhibited antimicrobial activity against biofilms formed by *C. albicans*, *S. mutans*, and *E. faecalis*.

#### 3.2.3. Side Effects


*Cytotoxicity of SeNPs*


As shown in Table 3, the toxicity level of SeNPs was evaluated in twelve studies. The cytotoxicity of SeNPs was determined through the percentage of cell viability, half-maximal inhibitory concentration (IC_50_), lethal concentration 50 (LC_50_), and 50% cytotoxic concentration (CC_50_). In seven articles, the biogenic SeNPs were reported to be non-toxic to normal human cells. Only one study showed that biologically synthesized SeNPs displayed a high cytotoxicity against Vero cells. Moreover, five studies evaluated the cytotoxicity of synthetic SeNPs, which were non-toxic at low doses. Most studies demonstrated that the percentage of cell viability increased as the SeNP concentration decreased. Interestingly, three studies also investigated the anticancer activity of biogenic SeNPs against various human cancer cell lines, obtaining promising results in a concentration-based manner.


*Antioxidant activity of SeNPs*


The antioxidant activity of SeNPs was also evaluated to determine if they possess potential benefits or adverse effects when they act as antioxidants or prooxidants. To address this aspect, the antioxidant activity of SeNPs was assessed in eight studies using ABTS and DPPH assays, as detailed in Table 4. In seven studies, the antioxidant activity was more effective as the SeNP concentration increased. In one study, the antioxidant activity was effective at a lower concentration than the control. In all the studies, the potent antioxidant activity was effectively demonstrated with biologically synthesized SeNPs.

## 4. Discussion

This scoping review aimed to investigate the potential role of SeNPs in combating oral infections caused by various microbes. Following a rigorous selection process, a total of 27 studies were deemed suitable for inclusion in the review. These studies reported relevant in vitro antimicrobial efficacy, effective antioxidant activity, and limited toxicity.

### 4.1. Antimicrobial Efficacy of SeNPs against Pathogenic Oral Microorganisms

The antimicrobial activity of SeNPs was only reported in four (*C. albicans*, *S. mutans*, *E. faecalis*, and *P. gingivalis*) out of nine oral microorganisms included in the search strategy. The reported data are representative of the therapeutic potential of SeNPs, given that these four microorganisms studied are linked to pathogenic oral conditions. These microorganisms are associated with various oral manifestations, such as dental caries, mucosal lesions in immunocompromised patients, superinfections in periodontitis, and persistent root canal infections [3,4,5,21]. The prevalence and incidence of all forms of oral candidiasis have increased in recent decades, and nystatin is considered one of the main recommended treatments for such cases [5,69]. In this regard, the article published by Nile et al. [62] contributes to the evidence of the ability of SeNPs to act as carriers of nystatin for improved stability, localized delivery, and sustained release. This nanoconjugate achieved a remarkable synergistic antimicrobial effect, which could help to reduce the emergence of microbial resistance [70]. Thus, this opens new opportunities for the use of SeNPs as a vehicle to deliver antimicrobial molecules and an effective strategy to combat oral infections.

Overall, all the included articles demonstrated that the antimicrobial activity of SeNPs against planktonic microorganisms is dependent on both the concentration of the NPs and the duration of the treatment. Previously reported studies support these observations [71,72,73]. In particular, the fact that susceptibility increased when SeNP concentration increased may be due to the presence of high amounts of Se ions, which is directly related to an increase in the stress levels of the microbial cells, thereby leading to their death [37]. The strong antimicrobial effects of SeNPs are mainly associated with DNA damage, protein degradation, and cell destruction [39]. This characteristic is also directly influenced by the microbial species as a result of their distinct cell surface attributes [15]. The articles included in this review demonstrated that *C. albicans* was the most sensitive to SeNPs, followed by both Gram-positive strains and *P. gingivalis*. This may be attributed to the intricate design of the Gram-negative bacteria’s cell wall (the outer membrane) [74]. Additionally, Gram-positive bacteria have a higher surface negative charge than Gram-negative bacteria, which can attract NPs [56]. However, more evidence is needed to establish a real trend in the antimicrobial activity of SeNPs, considering that the methodology employed was highly varied, and the doses applied were not clear in some reports. In fact, it could be affirmed that the heterogeneity in the methods and the lack of clarity in many of the articles do not allow us to ensure the properties that were observed. The standardization of the methods would greatly improve our knowledge on this subject.

In regard to the antibiofilm activity of SeNPs, several observations must be highlighted. Firstly, Guisbiers et al. [67] demonstrated that the size of the produced SeNPs was a relevant key parameter in the inhibition of *C. albicans* biofilms. The smallest SeNPs (~50 nm) achieved the highest antibiofilm activity. This can be attributed to the fact that the smaller the size of the NPs, the larger the surface area, and consequently, they have much more facility to interact with the surface of bacterial cells and to be trapped inside the plasma membrane or form a pore, enabling translocation [75,76]. It has been described that, owing to their high surface-to-volume ratio, NPs acquire an effective transport phenomenon within the biofilm matrix, and therefore, the NPs’ size controls the initial penetration of NPs within the extracellular material [77].

Secondly, the evidence suggests that the antibiofilm activity of SeNPs also seems to be linked to the synthesis methods [55] and the stabilizing agents [68]. The paper presented by Cremonini et al. [55] showed that biogenic SeNPs possess a better antimicrobial activity than chemically synthesized ones. According to several authors, increased antibiofilm activity can be attributed to the unique biogenic surface coating of biosynthesized SeNPs, which maintains the electrostatic stability needed to interact with the biofilm matrix components [55,77,78,79]. Moreover, the chitosan-stabilized SeNPs were very effective [68], which is consistent with observations that chitosan alone has significant action against Gram-positive bacterial and fungal biofilms [80,81]. Importantly, the morphology of NPs, which also plays an important role in their effectiveness against microbial species, strongly depends on the choice of stabilizers and synthesis methods [56]. The antimicrobial activity based on shape depends on how good the interaction is between the nanomaterials and biological entities [56]. The articles included in this scoping review demonstrated that spherical SeNPs displayed a high antimicrobial potency against various oral microorganisms, which could be associated with their closer contact with the bacteria surface [82]. According to previous research, SeNPs with a spherical and cubic shape have been used to enhance antitumor, antioxidant, and antimicrobial activities [65,83], whereas Se nanorods have been used in electrochemical sensors [84].

Lastly, Shahmoradi et al. [59] reported that the combination of aPDT and SeNPs caused a significant decrease in the viability of *E. faecalis* biofilms. aPDT is particularly interesting in the dental field as it has been shown to increase the antimicrobial activity of certain NPs [85]. Silver and chitosan nanoparticles have also been reported with good results when combined with aPDT against biofilms of *S. mutans* and *P. aeruginosa*/*S. aureus*, respectively [86,87]. Interestingly, it has even been seen that the combination of SeNPs with aPDT can deeply disinfect the dentinal tubules, reaching 400 nm in depth [59]. By contrast, aPDT alone could not produce an antibacterial effect at the same depth, but rather in superficial layers [59]. This is relevant since residual bacteria inside the root canal system are the cause of the appearance of persistent infections.

### 4.2. In Vitro Side Effects of SeNPs

The toxicity of SeNPs was assessed in various cell lines, revealing minimal or negligible toxicity towards normal cells, even at the same concentration range used for antimicrobial testing. Remarkably, SeNPs exhibited a robust antiproliferative effect on cancer cells [50]. The anticancer activity of SeNPs can be explained by their pro-oxidant property. The SeNPs produced oxidative stress, releasing ROS and resulting in DNA damage-induced cell death due to caspase-dependent apoptosis activation through mitochondrial cytochrome c release [34,88]. Interestingly, Cremonini et al. [55] showed that SeNPs were unable to cause damage to the immune system and non-immune cells or to induce the secretion of pro-inflammatory cytokines. From the data reported, one may conclude that SeNPs can exhibit different cytotoxicity effects depending on the stabilizers employed. In fact, the study conducted by Filipovic et al. [56] demonstrated that the formulation of SeNPs-BSA exhibited significantly lower cytotoxicity compared to the other two formulations tested (SeNPs-Chit and SeNPs-Gluc). Based on prior knowledge, BSA has recently received a lot of interest in the nanotechnology and pharmaceutical industries of late due to its biological safety, biodegradability, hydrophilicity, non-toxicity, and non-immunogenic properties [89,90]. Moreover, in accordance with previous research [91], the results of the DPPH and ABTS assays demonstrated that biogenic SeNPs have a strong ability to scavenge free radicals and prevent DNA oxidation in a concentration-dependent manner. As supported by the existing literature, the potent antioxidant activity of biogenic SeNPs is primarily associated with the high total phenolic content found in the organic cap surrounding these nanoparticles [45,92,93].

Based on the comprehensive findings of our study, it is evident that the synthesis method, structural and morphological characteristics, and the concentration of SeNPs play critical roles in determining their functionality and toxicity, particularly when considering their potential biomedical applications. In this regard, it is essential to evaluate the potential toxicity of the NPs with broad-spectrum bioactivities due to their ability to traverse different cell barriers, which could cause tissue injury [94,95].

### 4.3. Limitations

It is important to acknowledge certain limitations in our scoping review. Firstly, scoping reviews inherently have limitations in their scope, as they aim to offer a comprehensive overview rather than an in-depth analysis of a specific topic. As a result, a scoping review often does not undertake a meta-analysis. However, this approach was suitable, given that our goal was to map out the evidence on the antimicrobial activity of SeNPs against pathogenic oral microorganisms. Secondly, despite conducting a systematic review, it is possible that some studies might have been missed. Nevertheless, we implemented a sensitive search strategy, supplemented by a manual reference search, and employed a double independent review process to minimize this possibility. Additionally, a thorough assessment of the gray literature was conducted. Third, we limited the included studies to those disseminated in English, Spanish, or Portuguese. As such, our results could have missed important studies written in other languages. In addition, some limitations derive from the short horizon of the reviewed articles, all of which provide data obtained in vitro, which will be corrected when the products are tested in the complex physiological conditions generated in the oral cavity. Among the studies included in this review, only four potentially pathogenic oral microorganisms were used in the investigations, which might not completely reflect the antimicrobial effect of SeNPs in the mouth. In this regard, in vivo studies are highly required to support the efficacy of SeNPs and to evaluate their substantivity properties. Furthermore, an additional limitation of this study is that it focused on mono-species and mono-strain cultures without the inclusion of natural or artificial saliva as an additive. It is important to note that a typical oral-associated biofilm is composed of a highly intricate and diverse multi-species microbial community, embedded within an extracellular matrix of polysaccharides and glycoproteins. This complex biofilm structure can significantly influence the interactions and dynamics between nanoparticles and the microbial cells [96]. Therefore, to improve the clinical perspectives, there must be a better understanding of the effect of SeNPs on oral multispecies biofilms formed on biotic and abiotic surfaces.

### 4.4. Implications for Practice and Research

The use of SeNPs holds promising potential to mitigate the global impact of antibiotic resistance, which has emerged as a significant concern in recent decades. Dentistry certainly plays a significant role in contributing to the spread of antimicrobial resistance, and finding effective antimicrobial agents is crucial for improving treatment outcomes and patients’ quality of life. The promising in vitro results of SeNPs as antimicrobial agents in dentistry suggest their potential application in improving the success rate of dental treatments. By effectively combating microbial infections, SeNPs can enhance the efficacy of dental interventions, leading to better treatment outcomes and overall patient satisfaction. However, further studies are essential to critically assess the effectiveness of SeNPs in vivo and determine their real therapeutic impact. Future research should include more detailed information concerning SeNPs including concentrations employed, their physicochemical properties, and the experimental conditions to provide enough evidence to address the construction and development of well-designed and safe protocols.

## 5. Conclusions

Collectively, the results analyzed demonstrate that using SeNPs against pathogenic oral microorganisms appears effective in vitro in reducing planktonic and sessile microbial populations. At the same time, SeNPs seem able to induce cytotoxicity in human cancer cells with negligible effects on normal cells *in a dose-dependent manner.* The reviewed articles suggest that biogenic SeNPs are the most suitable candidates due to their antimicrobial and antioxidant efficacy and low cytotoxicity. Further research must focus on biofilm susceptibility assays under practical conditions, as well as on assessing the effectiveness of this alternative treatment in ex vivo and in vivo settings.

## Figures and Tables

**Figure 1 pharmaceutics-15-02253-f001:**
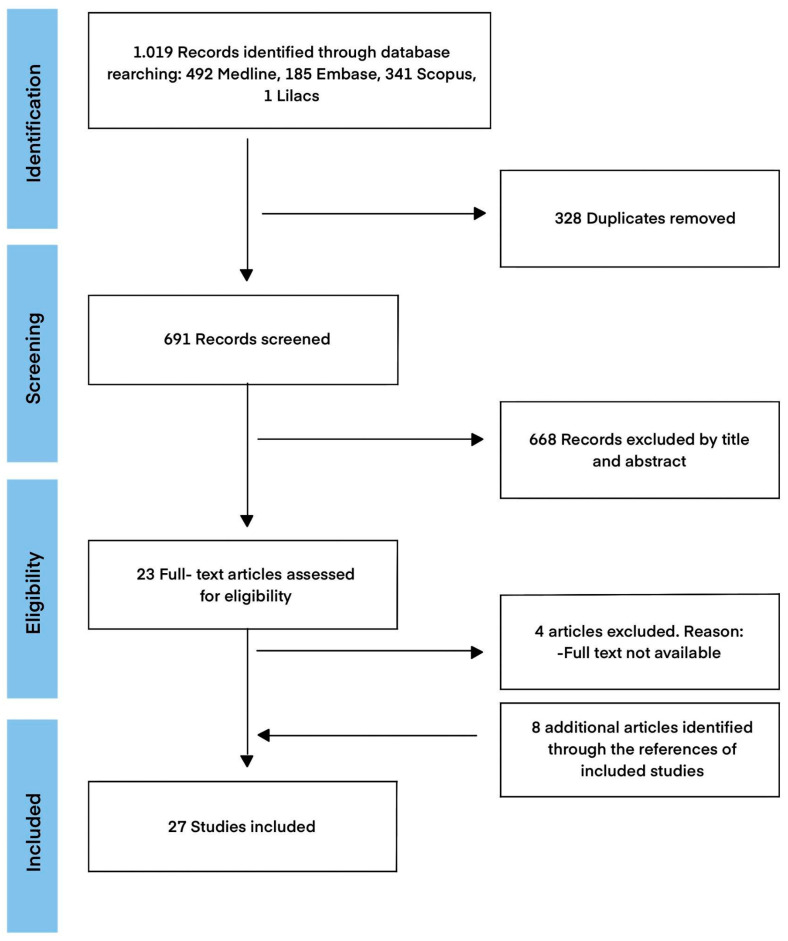
Flow diagram illustrating the process of selecting sources of evidence.

**Table 1 pharmaceutics-15-02253-t001:** Treatment efficacy data in a planktonic microbial organization.

Type of SeNPs	Microorganisms	Bacterial Incubation Time	Concentration of SeNPs (μg/mL)	Size of SeNPs (nm)	Efficacy	Main Conclusion	Reference
Biogenic	*C. albicans*	24 h	25–200	79.40 ± 44.26	MIC (μg/mL) = 25	SeNPs showed potent antifungal activity.	[38]
Biogenic	*C. albicans*	5 days	100–500	45–80	ZOI (mm) = 13.1 (100 μg/mL); 15.5 (200 μg/mL); 17.2 (300 μg/mL); 19.5 (400 μg/mL); 20.9 (500 μg/mL); Diniconazole (20 mg/mL) = 27.9 MIC (μg/mL) = 75	SeNPs exhibited antifungal activities, which increased in a concentration-dependent manner.	[45]
Biogenic	*E. faecalis*	ND	ND	29–195	ZOI (mm) = 32 ± 1 (50 μL); 35 ± 1 (100 μL); 36 ± 1 (150 μL)	SeNPs possessed antibacterial activity against *E. faecalis*.	[46]
Biogenic	*S. mutans*, *E. faecalis* and *C. albicans*	24–48 h	>5000	16–132	ZOI (mm) *S. mutans* = 10 (0.25 mg); 12 (0.5 mg); 15 (1 mg) *E. faecalis* = 8 (0.25 mg); 12 (0.5 mg); 20 (1 mg) *C. albicans* = 10 (0.25 mg); 24 (0.5 mg); 28 (1 mg) Ampicillin (5 mg) = 22 (*S. mutans*); 27 (*E. faecalis*) Cycloheximide (5 mg) = 34 (*C. albicans*)	SeNPs showed great potential as an oral antimicrobial agent.	[47]
Biogenic	*E. faecalis*	24 h	100–300	80–120	ZOI (mm) = 12.20 ± 0.63 (100 μg); 16.73 ± 0.27 (200 μg); 23.41 ± 0.50 (300 μg); Ampicillin (100 μg) = 21.16 ± 0.88 MIC (μg/mL) = 23.12 ± 1.89; Ampicillin = 10.41 ±1.06 MBC (μg/mL) = 52.21± 2.80; Ampicillin =18.56 ± 0.72	SeNPs exhibited antibacterial activity against *E. faecalis*.	[48]
Synthetic	*S. mutans* and *C. albicans*	24 h	5000	81.4	MIC (μg/mL) = 68 (*S. mutans*); 274 (*C. albicans*) MBC (μg/mL) *S. mutans* 274 (after 1–2 h); 137 (after 6–24 h) *C. albicans* not bactericidal effect (after 1–2 h); 274 (after 6–24 h)	Chit-SeNPs had significant antimicrobial activity against both *S. mutans* and *C. albicans*.	[49]
Biogenic	*C. albicans* and *S. mutans*	24–48 h	ND	ND	ZOI (mm) *C. albicans =* 30 (50 μL); 32 (100 μL); 35 (150 μL) *S. mutans =* 9 (50 μL); 11 (100 μL); 14 (150 μL)	SeNPs had antifungal and antibacterial activity.	[43]
Biogenic	*C. albicans*	3–5 days	15.62–2000	4–12.7	ZOI (mm) = 8.7 ± 0.1 (1.25 μg); 25.6 ± 0.7 (20 μg) MIC (μg/mL) = 125	SeNPs exhibited effective antimicrobial activity against unicellular fungi.	[50]
Biogenic	*C. albicans*	24–48 h	ND	ND	ZOI (mm) = 10 ± 1.8 (25 μL); 15 ± 2.1 (50 μL); 31 ± 3.3 (100 μL)	SeNPs had antimicrobial efficacy against *C. albicans*.	[44]
Synthetic	*E. faecalis*	24 h	5000	50–105	MIC (μg/mL) = 274 MBC (μg/mL) = not bactericidal effect (after 1–2 h); 274 (after 6–24 h)	Chit-SeNPs revealed good antimicrobial activity against Gram-positive bacteria.	[51]
Synthetic	*C. albicans*	48 h	10–200	80–220	MIC (μg/mL) = 70	SeNPs had good antifungal activity.	[52]
Biogenic	*C. albicans*	48 h	ND	5–25	Fungal growth inhibition (%) = 70.86	SeNPs had desirable antifungal properties.	[53]
Biogenic	*S. mutans*, *E. faecalis* and *C. albicans*	24 h	ND	10–25	ZOI (mm) *E. faecalis =* 12 (25 μL); 12 (50 μL); 12 (100 μL) *S. mutans =* 8 (25 μL); 13 (50 μL); 15 (100 μL) *C. albicans =* 8 (25 μL); 8 (50 μL); 8 (100 μL)	SeNPs displayed antimicrobial activity against oral pathogens. The maximum inhibitory effect was observed against *S. mutans* at the volume of 100 μL.	[54]
Biogenic and Synthetic	*C. albicans*	ND	50–500	160.6 ± 52.24	MIC (μg/mL) Sm-SeNPs = 256 Bm-SeNPs = 512 Ch-SeNPs = >512	Neither the biogenic nor the synthetic SeNPs did not inhibit the growth of planktonic *C. albicans* strains.	[55]
Synthetic	*E. faecalis* and *C. albicans*	24 h	400	100–200	MIC (μg/mL) *C. albicans* SeNPs-BSA & SeNPs-Chit =25 *E. faecalis* SeNPs-Gluc =72 SeNPs-BSA & SeNPs-Chit =100 SeNPs-Gluc =290	SeNPs-BSA and SeNPs-Chit showed higher antimicrobial activity than SeNPs-Gluc, for both *E. faecalis* and *C. albicans*.	[56]
Biogenic	*E. faecalis*	48 h	1000	40–150	MIC (µg/mL) = 25; 17 (Gentamicin) ZOI (mm) = 1.33 (10 μg); 16.50 (20 μg); 21 (30 μg); 28.50 (40 μg).	SeNPs had potential as an effective antimicrobial agent for root canal disinfection.	[57]
Synthetic	*S. mutans*	24 h	64–128	77 ± 27	VCC LED + MB + SeNPs 128 μg/mL = no bacterial reduction LED + MB + SeNPs 64 μg/mL = no bacterial reduction SeNPs 128 μg/mL = no bacterial reduction SeNPs 64 μg/mL = no bacterial reduction LED + MB = no bacterial reduction LED = no bacterial reduction	The combination of LED + MB + SeNPs and SeNPs alone did not significantly reduce the number of planktonic *S. mutans* compared to the control group.	[58]
Synthetic	*E. faecalis*	24 h	64–128	77 ± 27	VCC LED + MB + SeNPs 128 μg/mL = Log 2 CFU/mL reduction LED + MB + SeNPs 64 μg/mL = Log 1 CFU/mL reduction LED + MB = Log 1 CFU/mL reduction LED = Log 0 CFU/mL reduction	LED + MB + SeNPs 128 μg/mL showed a slight reduction in planktonic *E. faecalis.*	[59]
Biogenic	*S. mutans* and *C. albicans*	24 h	ND	30–200	ZOI (mm) *S. mutans =* 25 (50 μL); 30 (100 μL); 33 (150μL); 15 (Amoxicillin) *C. albicans =* 9 (50 μL); 10 (100 μL); 11 (150μL); 12 (Fluconazole)	SeNPs showed high antimicrobial activity against *C. albicans* and *S. mutans*.	[60]
Synthetic	*P. gingivalis*	96 h	2–2048	70	VCC 2048 μg/mL = Log 3 CFU/mL reduction 1024 μg/mL = Log 1 CFU/mL reduction <1024 μg/mL = Log 0 CFU/mL reduction MBC (μg/mL): 2048 μg/mL = not bactericidal effect	SeNPs showed an inhibitory effect against *P. gingivalis*, which is concentration-dependent. The highest concentration of SeNPs was unable to kill *P. gingivalis* bacteria for a fixed period of time.	[61]
Biogenic	*C. albicans*	48 h	3.9–500	220–242	Fungal growth inhibition (%) = SeNPs = 3 (3.9 μg/mL); 5 (15.62 μg/mL); 20 (62.50 μg/mL); 30 (125 μg/mL); 45 (500 μg/mL) SeNP@PVP-Nystatin = 60 (3.9 μg/mL); 70 (15.62 μg/mL); 80 (62.50 μg/mL); 85 (125 μg/mL); 100 (500 μg/mL)	The biogenic nanoconjugate SeNP@PVP-Nystatin inhibited the growth of *C. albicans*.	[62]
Biogenic	*S. mutans* and *C. albicans*	3–5 days	2000	14.5	ZOI (mm) *S. mutans* = 54 ± 1.48 *C. albicans* = 41 ± 0.70 MIC (μg/mL) *S. mutans* = 3.9 *C. albicans* = 15.62	SeNPs exhibited antimicrobial efficacy against *C. albicans and* *S. mutans.*	[63]
Biogenic	*C. albicans*	24 h	10–100	64–93	ZOI (mm) SeNPs = 8 (10 μg/mL); 12 (50 μg/mL); 11 (100 μg/mL) SeNPs + plant extract (*Camellia sinensis*) = 12 (10 μg/mL); 14 (50 μg/mL); 10 (100 μg/mL) Ampicillin = 9 (10 μg/mL); 11 (50 μg/mL); 6 (100 μg/mL)	SeNPs possessed antimicrobial potential, being higher when used in combination with the plant extract.	[64]
Biogenic	*C. albicans*	24 h	50–400	25–75	ZOI (mm) = 8 (50 μg/mL); 10 (100 μg/mL); 12(200 μg/mL); 15(300 μg/mL); 18 (400 μg/mL)	SeNPs exhibited high activity against *C. albicans.*	[65]
Biogenic	*C. albicans*	5 h	25–100	ND	Growth inhibition (%) = 75 (100 μg/mL); Ciprofloxacin (100 μg/mL) = 65	SeNPs showed antimicrobial activity against *C. albicans*.	[66]

MIC, minimal inhibitory concentration; MBC, minimal bactericidal concentration; ZOI, zone of inhibition; CFU, colony-forming units; BS, bacterial survival; MB, methylene blue; BSA, bovine serum albumin; Chit, chitosan; Gluc, glucose; LED, light-emitting diode; Sm, *Stenotrophomonas maltophilia*; Bm, *Bacillus mycoides*; Ch, chemically synthesized; VCC, viable cell counts; ND, not determined.

**Table 2 pharmaceutics-15-02253-t002:** Treatment efficacy data in biofilm microbial organization.

Type of SeNPs	Microorganisms	Bacterial Incubation Time	Concentration of SeNPs (μg/mL)	Size of SeNPs (nm)	Efficacy	Main Conclusion	Reference
Synthetic	*C. albicans*	24 h	5–25	50–400	BI (%) SeNPs (just after their synthesis) = 10 (5 μg/mL); 20 (15 μg/mL); 30 (25 μg/mL) SeNPs (with the smallest size) = 15 (5 μg/mL); 40 (15 μg/mL); 50 (25 μg/mL) SeNPs (with crystalline structure) = 10 (5 μg/mL); 30 (15 μg/mL); 50 (25 μg/mL)	SeNPs seemed to be a good candidate as antifungal agents. The size and crystallinity of the produced SeNPs are key parameters for inhibiting *C. albicans* biofilm.	[67]
Synthetic	*C. albicans*	24 h	0.05–2500	96	BI (%) SeNPs 25 μg/mL = 59 ± 7 Chit 2500 μg/mL + SeNPs 25 μg/mL = 80 ± 2	Chit-SeNPs showed the most potent inhibition against *C. albicans* biofilms.	[68]
Biogenic and Synthetic	*C. albicans*	ND	50–500	160.6 ± 52.24	BI (%) Sm-SeNPs = 61 ± 0.5 (50 μg/mL); 60 ± 1 (250 μg/mL; 94 ± 1 (500 μg/mL) Bm-SeNPs = 60 ± 6.5 (50 μg/mL); 74 ± 2.5 (250 μg/mL); 93 ± 0.5 (500 μg/mL) Ch-SeNPs = No biofilm inhibition (50 μg/mL); No biofilm inhibition (250 μg/mL); 9 ± 0.7 (500 μg/mL)	Biogenic SeNPs were potentially suitable as antimicrobial agents for *C. albicans* biofilm.	[55]
Synthetic	*C. albicans*	24	400	100–200	BI (%) = 96 ± 4.2	SeNPs displayed significant inhibition on *C. albicans* biofilms.	[56]
Biogenic	*E. faecalis*	48 h	1000	40–150	BI (%) = 65 VCC (%) = 21.38 (24 h); 12.13 (48 h) VCC (%) Ca(OH)_2_ =72.20–58.10; CHX = 30.03–19.15; NaOCl = 27.09–17 (24–48 h, respectively)	SeNPs demonstrated their potential to be a root canal disinfectant combating bacterial biofilm.	[57]
Synthetic	*E. faecalis*	24 h	64–128	77 ± 27	VCC LED + MB + SeNPs 128 μg/mL = Log 2 CFU/mL reduction LED + MB + SeNPs 64 μg/mL = Log 2 CFU/mL reduction LED + MB = Log 1 CFU/mL reduction LED = Log 1 CFU/mL reduction	SeNPs could promote aPDT efficiency and provide appropriate antibacterial properties against *E. faecalis* biofilms.	[59]
Synthetic	*S. mutans*	24 h	64–128	77 ± 27	VCC LED + MB + SeNPs 128 μg/mL = no bacterial reduction LED + MB + SeNPs 64 μg/mL = no bacterial reduction SeNPs 128 μg/mL = no bacterial reduction SeNPs 64 μg/mL = no bacterial reduction LED + MB = no bacterial reduction LED = no bacterial reduction	SeNPs did not enhance aPDT activity or provide a considerable antibiofilm effect against *S. mutans*.	[58]
Biogenic	*C. albicans*	48 h	3.9–500	220–242 nm	BI (%) SeNPs = 5 (3.9 μg/mL); 10 (15.62 μg/mL); 20 (62.50 μg/mL); 25 (125 μg/mL); 40 (500 μg/mL) SeNP@PVP-Nystatin = 50 (3.9 μg/mL); 70 (15.62 μg/mL); 80 (62.50 μg/mL); 90 (125 μg/mL); 100 (500 μg/mL)	The biogenic nanoconjugate SeNP@PVP-Nystatin showed inhibition against *C. albicans* biofilms as concentration increased.	[62]

BI, biofilm inhibition; CFU, colony-forming units; MB, methylene blue; Chit, chitosan; Sm, *Stenotrophomonas maltophilia*; Bm, *Bacillus mycoides*; Ch, chemically synthesized; VCC, viable cell counts; Ca (OH)_2_, calcium hydroxide; CHX, 2% chlorhexidine; NaOCl, 5.25% sodium hypochlorite; ND, not determined.

**Table 3 pharmaceutics-15-02253-t003:** Side effects in the in vitro studies.

Type of SeNPs	Cell Line	Time	Side Effects	Conclusion	Reference
Biogenic	- THLE2 Normal liver cells - Vero Normal kidney cells	48 h	IC_50_ (μg/mL) Kidney cells = 233.08 Liver cells = 849.21	Biogenic SeNPs were less toxic in normal liver cells and much safer than in normal kidney cells.	[38]
Biogenic	BSLA	48 h	LC_50_ (μg/mL) = 20	SeNPs were safe and exhibited limited toxicity.	[47]
Biogenic and Synthetic	- Human primary fibroblast CCD1112Sk cells (ATCC CRL-2429) - Dendritic cells	24 h	Cell viability (%) Dendritic cells: Sm-SeNPs = 98 (50 μg/mL); 98 (100 μg/mL); 90 (500 μg/mL) Bm-SeNPs = 98 (50 μg/mL); 95 (500 μg/mL) Ch-SeNPs = 100 (50 μg/mL); 95 (500 μg/mL) Fibroblasts cells: Sm-SeNPs = 100 (50 μg/mL); 100 (500 μg/mL) Bm-SeNPs = 100 (50 μg/mL); 100 (500 μg/mL) Ch-SeNPs = 100 (50 μg/mL); 100 (500 μg/mL)	- Biogenic and synthetic SeNPs did not affect human dendritic cell or fibroblast viability. - The biogenic and synthetic SeNPs did not stimulate the secretion of proinflammatory and immunostimulatory cytokines at low concentrations.	[55]
Synthetic	- MRC-5 cells	24 h	Cell viability (%) SeNPs-BSA = 70 (400 μg/mL); 100 (20 μg/mL); 90 (1 μg/mL); 100 (0.1 μg/mL) SeNPs-Chit = N/A (400 μg/mL); 90 (20 μg/mL); 90 (1 μg/mL); 90 (0.1 μg/mL) SeNPs-Gluc = 18 (400 μg/mL); 70 (20 μg/mL); 85 (1 μg/mL); 65 (0.1 μg/mL)	- SeNPs-BSA, SeNPs-Chit and SeNPs-Gluc had no cytotoxicity effects at lower doses. - Cell survival after treatment was higher for SeNPs-BSA.	[56]
Biogenic	- Vero cell line (ATCC-CCL-81) - Prostate cancer cell line (PC3)	ND	IC_50_ (μg/mL) Vero cell line = 316.73 PC3 cells = 99.25	- SeNPs were non-toxic to normal human cell lines (Vero). - SeNPs had anticancer activity against PC3 cells.	[50]
Synthetic	- Human retinal pigment epithelial cell line ARPE-19	24 h	CC_50_ (μg/mL) = 26.3	- SeNPs were non-toxic at lower doses.	[68]
Synthetic	- Human fibroblast cells	24 h	Cell viability (%) = 85 (4 μg/mL); 78 (16 μg/mL); 55 (64 μg/mL); 50 (128 μg/mL)	Human fibroblast cells had increased survival at low SeNPs concentrations.	[59]
Synthetic	- MC3T3 cells	3,5,7 days	Cell viability (Absorbance 450/620 nm) Day 3: Control (0 μg/mL) =1.8 2 μg/mL = 1.7 64 μg/mL = 1.6 2048 μg/mL = 1.8 Day 5: Control (0 μg/mL) =1.9 2 μg/mL =1.9 64 μg/mL = 1.7 2048 μg/mL = 1.8 Day 7: Control (0 μg/mL) =1.8 2 μg/mL =1.8 64 μg/mL = 1.4 2048 μg/mL = 1.3	SeNPs did not show significant toxicity on days 3, 5, or 7 at any concentration tested.	[61]
Biogenic	- Human embryonic kidney 293 cell line (HEK-293)	24 h	HEK Cell Growth (%) = 99 (3.9 μg/mL); 99 (31.25 μg/mL); 90 (62.50 μg/mL)	The biogenic nanoconjugate SeNP@PVP-Nystatin was not cytotoxic at concentrations lower than 125 μg/mL in HEK-293 cells.	[62]
Biogenic	- Vero normal cell line - Human breast cancer cell line (MCF7) - Human osteosarcoma cell line (MG-63)	ND	IC_50_ (μg/mL) Vero cell line = 113.73 MCF7 = 69.8 MG-63 = 47.9 Vero viability (%) = 90 (31.25 μg/mL); 45 (125 μg/mL); 10 (500 μg/mL); 5 (1000 μg/mL) Cell proliferation inhibition (%) MCF7 = 15 (31.25 μg/mL); 80 (125 μg/mL); 92 (500 μg/mL); 95 (1000 μg/mL) MG-63 = 30 (31.25 μg/mL); 88 (125 μg/mL); 95 (500 μg/mL); 98 (1000 μg/mL)	- SeNPs were non-toxic for human normal cell lines at lower concentrations. - SeNPs exhibited anticancer activity against both cancerous cell lines.	[63]
Biogenic	- Vero normal cell line	24 h	IC_50_ (μg/mL) = 1.02 ± 0.8 Cell viability (%) = 20.79 (12.5 μg/mL)	SeNPs demonstrated a high cytotoxicity against Vero cells.	[64]
Biogenic	- Human breast cancer cell line (MCF7) - Prostate cancer cell line (PC3) - Vero Normal kidney cells - Human lung fibroblast cell line (WI38)	24 h	Cell viability (%) MCF7 = 99.6 ± 3.2 (125 μg/mL); 26.3 ± 1.8 (500 μg/mL) PC3 = 89.7 ± 0.9 (125 μg/mL); 8.3 ± 0.9 (500 μg/mL) Vero = 98.4 ± 3.1 (125 μg/mL); 44.4 ± 0.7 (500 μg/mL) WI38 = 99.9 ± 1.2 (125 μg/mL); 43.1 ± 0.9 (500 μg/mL) IC_50_ (μg/mL) MCF7 = 283.8± 7.5 PC3 = 225.7 ± 3.6 Vero = 472.8 ± 5.8 WI38 = 454.8 ±29.9	- At high concentrations (≥125 μg/mL) of SeNPs, the viability was highly decreased for cancerous cells compared to the viability of normal cells. - The toxicity of biogenic SeNPs was the highest against PC3 compared to MCF7. - The biocompatibility of both normal cell lines toward various concentrations of SeNPs was similar.	[65]

IC_50_, half-maximal inhibitory concentration; LC_50_, lethal concentration 50; CC_50_, the 50% cytotoxic concentration; BSA, bovine serum albumin; Chit, chitosan; Gluc, glucose; BSLA, brine shrimp lethality assay; Sm, *Stenotrophomonas maltophilia*; Bm, *Bacillus mycoides*; Ch, chemically synthesized; ND, not determined.

**Table 4 pharmaceutics-15-02253-t004:** Antioxidant activity of SeNPs.

Type of SeNPs	Antioxidant Assay	Antioxidant Activity	Conclusion	Reference
Biogenic	ABTS assay DPPH assay	ABTS and DPPH radical inhibition (%) = 93 and 90, respectively (Control); 92 and 89, respectively (500 μg/mL SeNPs)	SeNPs significantly scavenged the ABTS and DPPH radicals. The antioxidant activity of SeNPs increased in a concentration-dependent manner.	[45]
Biogenic	DPPH assay	DPPH radical inhibition (%) = 93.15 (50 μg/mL of SeNPs)	SeNPs showed potent antioxidant activity.	[47]
Biogenic	ABTS assay DPPH assay	DPPH assay—IC_50_ (μg/mL) = 58.98 ± 0.70 ABTS assay—IC_50_ (μg/mL) = 66.10 ± 1.01	SeNPs had a dose-dependent antioxidant effect.	[48]
Biogenic	DPPH assay	Antioxidant activity (%) = 50 (30 μg/mL of SeNPs)	SeNPs had strong antioxidant activity.	[50]
Biogenic	DPPH assay	SeNPs DPPH radical inhibition (%) = 55 (10 μg/mL), 60 (20 μg/mL), 65 (30 μg/mL), 68 (40 μg/mL), 70 (50 μg/mL) BHT (Control) DPPH radical inhibition (%) = 45 (10 μg/mL), 55 (20 μg/mL), 68 (30 μg/mL), 80 (40 μg/mL), 90 (50 μg/mL)	SeNPs displayed effective antioxidant activity.	[54]
Biogenic	DPPH assay	DPPH radical inhibition (%) = 50 (31.25 μg/mL); 75 (125 μg/mL); 95 (500 μg/mL); 100 (1000 μg/mL) DPPH assay—IC_50_ (μg/mL) = 27	SeNPs exhibited strong antioxidant activity.	[63]
Biogenic	DPPH assay	DPPH radical inhibition (%) = 5 (100 μg/mL); 10 (300 μg/mL); 38.6 (500 μg/mL)	The antioxidant activity of SeNPs increased as concentration increased.	[64]
Biogenic	DPPH assay	DPPH radical inhibition (%) = 19.3 ± 4.5 (1.95 μg/mL); (86.6 ± 0.6 (1000 μg/mL) DPPH assay—EC_50_ (μg/mL) = 28.7 ± 1.6	SeNPs possessed a high antioxidant activity.	[65]

IC_50_, half-maximal inhibitory concentration; EC_50_, half-maximal effective concentration; BHT, butylated hydroxytoluene.

## Data Availability

Not applicable.

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
