# Peer review of "Antimicrobial Activity of Selenium Nanoparticles (SeNPs) against Potentially Pathogenic Oral Microorganisms: A Scoping Review"

_pharmaceutics, 2023, doi:10.3390/pharmaceutics15092253_

Round 1
Reviewer 1 Report
Authors have list all important parameters but there is missing connection among those parameters (Microorganisms, Type of SeNPs, Bacterial Incubation time, Concentration of SeNPs (μg/mL), Size of SeNPs (nm)) with Efficacy upon pathogenic oral microorganisms.
Authors need to do data analysis and illustrate the relationship with graphics.
Which microbial were the most studied?
How many type of SENPs were the most studied?
What the incubation time were the most studied?
Which size of SeNPs is the most effect upon pathogenic oral microorganisms?
What concentration of SeNPs range is the most effect?
Author Response
Reviewer #1,
1- Authors have listed all important parameters but there is missing connection among those parameters (Microorganisms, Type of SeNPs, Bacterial Incubation time, Concentration of SeNPs (μg/mL), Size of SeNPs (nm)) with Efficacy upon pathogenic oral microorganisms.
Response: We thank the reviewer for his/her thoughtful and thorough review and believe his/her input has been invaluable to make our manuscript of higher quality. We have done our best to connect all the parameters indicated above with the efficacy of SeNPs against pathogenic oral microorganisms.
2- Authors need to do data analysis and illustrate the relationship with graphics. Which microbial were the most studied?
How many types of SENPs were the most studied?; What the incubation time were the most studied?; Which size of SeNPs is the most effect upon pathogenic oral microorganisms?; What concentration of SeNPs range is the most effect?
Response:
- Which microbial were the most studied? C.albicans (Please see lines 182-185; 189-190).
- How many types of SENPs were the most studied? The most studied SeNPs were the biogenic ones (please see lines 190-192) with sizes lower than 100 nm (please see lines 194-197) and spherical shape.
- What the incubation time were the most studied? As can be seen in Tables 1 & 2, the most studies incubation time was 24 h, followed by 48 h.
- Which size of SeNPs is the most effect upon pathogenic oral microorganisms? The smaller the size, the higher the antimicrobial effect for all pathogenic strains (please, see Tables 1 & 2, and also lines 402-410 in discussion section). Therefore, the smallest SeNPs reported in this review possess high antimicrobial activity.
- What concentration of SeNPs range is the most effect? SeNPs reported have most been tested at concentrations ≤ 500 μg/mL (please see lines 197-200). As can be seen in Tables 1 & 2, the antimicrobial activity of SeNPs increased in a concentration-dependent manner.

Reviewer 2 Report
The manuscript from Sans-Serramitjana et al. synthesizes the antimicrobial applications of selenium nanoparticles to combat microbial pathogens causing oral infectious diseases. The scoping review is properly conducted: the authors identify the research team and the research question, indicate the sources of information and define the inclusion and exclusion criteria as well as the data charting process. The manuscript is well-written and merits publication.
Comments:
1. One of the results obtained by the authors is that the antimicrobial activity of SeNPs against planktonic microorganisms is nanoparticle concentration and treatment time-dependent (lines 385-386). Besides size and surface chemistry, particle form is a crucial parameter determining the interaction of nanomaterials with biological entities. Biogenic SeNPs are spherical, but synthetic nanoparticles can be rod-like. Therefore, the authors should provide information about the effect of particle shape on the antimicrobial activity of selenium nanoparticles.
2. Lines 405-408: The authors ascribe the highest antibiofilm activity of the smallest SeNPs to translocation of the nanomaterial across the cell wall and plasma membrane due to a larger surface area. For me, it is not clear how increasing the nanoparticle surface area would facilitate nanomaterial translocation. Please, explain.
On the other hand, biofilms consist of microbial communities embedded in an extracellular matrix composed of polysaccharides, proteins and extracellular DNA. In my opinion, providing information about the interaction of SeNPs with the extracellular matrix would improve the quality of the manuscript.
3. Lines 430 and 431: The authors correlate the anticancer activity of SeNPs to its binding to DNA and serum proteins. As it stands, the relationship between anticancer activity and binding to serum proteins is not clear. Please, clarify.
4. Fig. 1 looks blurred, please improve its quality
Author Response
Reviewer #2,
The manuscript from Sans-Serramitjana et al. synthesizes the antimicrobial applications of selenium nanoparticles to combat microbial pathogens causing oral infectious diseases. The scoping review is properly conducted: the authors identify the research team and the research question, indicate the sources of information and define the inclusion and exclusion criteria as well as the data charting process. The manuscript is well-written and merits publication.
Response: We would like to thank the reviewer for taking the necessary time and effort to revise this manuscript. We sincerely appreciate all your valuable comments and suggestions, which helped us in improving the quality of the manuscript.
1. One of the results obtained by the authors is that the antimicrobial activity of SeNPs against planktonic microorganisms is nanoparticle concentration and treatment time-dependent (lines 385-386). Besides size and surface chemistry, particle form is a crucial parameter determining the interaction of nanomaterials with biological entities. Biogenic SeNPs are spherical, but synthetic nanoparticles can be rod-like. Therefore, the authors should provide information about the effect of particle shape on the antimicrobial activity of selenium nanoparticles.
Response: We have made your recommended changes. The text below has been included in the manuscript within the discussion section (lines 419-428). Further, please see lines 196-197 (results section).
Importantly, the morphology of NPs, which also plays an important role in their effectiveness against microbial species, strongly depends on the choice of stabilizers and synthesis methods [56]. The antimicrobial activity based on shape depends on how good the interaction is between the nanomaterials and biological entities [56]. The articles included in this scoping review demonstrated that spherical SeNPs displayed a high antimicrobial potency against various oral microorganisms, which could be associated with their closer contact with the bacteria surface [82]. According to previous research, SeNPs with spherical and cubic shape have been used to enhance antitumor, antioxidant, and antimicrobial activities [65,83], whereas Se nanorods have been used in electrochemical sensors [84].
2. Lines 405-408: The authors ascribe the highest antibiofilm activity of the smallest SeNPs to translocation of the nanomaterial across the cell wall and plasma membrane due to a larger surface area. For me, it is not clear how increasing the nanoparticle surface area would facilitate nanomaterial translocation. Please, explain.
On the other hand, biofilms consist of microbial communities embedded in an extracellular matrix composed of polysaccharides, proteins and extracellular DNA. In my opinion, providing information about the interaction of SeNPs with the extracellular matrix would improve the quality of the manuscript.
Response: We have clearly explained how increasing the nanoparticle surface area would facilitate nanomaterial translocation and how SeNPs are able to interact with the extracellular matrix. The text below has been added to the manuscript within the discussion section (lines 455-457):
This can be attributed to the fact that the smaller the size of the NPs, the larger the surface area, and consequently, they have much more facility to interact with the surface of bacterial cells and to be trapped inside the plasma membrane or form a pore, enabling translocation [75,76]. It has been described that, owing to their high surface-to-volume ratio, NPs acquire an effective transport phenomenon within the biofilm matrix, and therefore, the NPs size controls the initial penetration of NPs within the extracellular material [77].
3. Lines 430 and 431: The authors correlate the anticancer activity of SeNPs to its binding to DNA and serum proteins. As it stands, the relationship between anticancer activity and binding to serum proteins is not clear. Please, clarify.
Response: Thank you for your valuable feedback. We agree with the reviewer that although there is evidence of the serum proteins-binding properties of SeNPs, but the relationship between anticancer activity and SeNPs-serum proteins interaction is not clear yet. However, considering that nanoparticles can enter the mitochondria and induce conformational changes of CytC (who switchs on the apoptotic pathways), one may think that serum proteins are linked indirectly to the anticancer activity exerted by SeNPs.
We have improved the expression related to this fact in order to clarify this concept and avoid confusion. Please see lines 443-446 (discussion section).
4. Fig. 1 looks blurred, please improve its quality
Response: Sorry for this mistake. It has been corrected.

Reviewer 3 Report
A very strange article. Poorly structured. The title of the article does not match the content. The necessity and significance of this work is not completely clear.
Practically not covered in the analysis of the work of 2023.
One of the output phrases is At the same time, SeNPs seem unable to cause significant damage to human cells. There are convincing results of both prooxidant and antioxidant effects of selenium nanoparticles. https://pubmed.ncbi.nlm.nih.gov/34639150/ Aren't cancer cells human cells?
The English language needs serious improvement. Lots of dificult sentences.
Author Response
Reviewer #3,
1. A very strange article. Poorly structured. The title of the article does not match the content. The necessity and significance of this work is not completely clear.
Response: We thank the reviewer for his/her thoughtful and thorough review and believe his/her input has been invaluable to make our manuscript of higher quality. The article has carefully revised and modified according to the suggestions received in order to improve its structure and clarify the significance of this work.
2. Practically not covered in the analysis of the work of 2023.
Response: We appreciate your comment. As per your recommendation, we have conducted a search on the articles published in 2023 following the methodology described in the manuscript. A total of 5 articles have been added to the manuscript (please see references 62,63, 65, 65, 66). We firmly believe that the incorporation of the analysis of the work of 2023 provides improved robustness to this scoping review.
3. One of the output phrases is At the same time, SeNPs seem unable to cause significant damage to human cells. There are convincing results of both prooxidant and antioxidant effects of selenium nanoparticles. https://pubmed.ncbi.nlm.nih.gov/34639150/ Aren't cancer cells human cells?
Response: We totally agree with your remarks. Some sentences have been modified to clarify the expression related to the cytotoxic effect of SeNPs. Please, see lines 515-516 (conclusions section). Moreover, the reference proposed has been added to the discussion section to justify the prooxidant-antioxidant effect of SeNPs depending on their concentration (please see lines 455-457).
4. The English language needs serious improvement. Lots of difficult sentences.
Response: We thank the reviewer for this valuable comment. We have polished this manuscript by MDPI Language Editing Services.

Round 2
Reviewer 1 Report
Authors have improved manuscript with figure 1 and tables clearly.
Please move abstract in the first page.
Method
Please move content of 3.1 to method part not in the result section
Line 179 to line 200 should be 4.6 cm from left margin.
Result
It is great to employ tables. Please move table closer to the sentence after the table was introduced.
Please introduce graphs to 3xplain tables.
Author Response
Reviewer #1,
1. Authors have improved manuscript with figure 1 and tables clearly.
Response: Thank you for your valuable feedback.
2. Please move abstract in the first page.
Response: It has been corrected (lines 25-44).
3. Please move content of 3.1 to method part not in the result section
Response: We appreciate your comment. However, at this point, we consider that we cannot make this change because of the following reason:
The manuscript adhered to the guideline of Preferred Reporting Items for Systematic Reviews and Meta-Analyses Extension for Scoping Reviews (PRISMA-ScR) following the editorial standards of Pharmaceutics. As per PRISMA-ScR, the Selection of Sources of Evidence should be reported in the results section. This section should provide the number of sources of evidence screened, the number of articles assessed for eligibility and included in the review, together with the reasons for exclusions at each stage, ideally using a flow diagram.
4. Line 179 to line 200 should be 4.6 cm from left margin.
Response: It has been corrected. The corrections are highlighted in the manuscript by track changes.
5. It is great to employ tables. Please move table closer to the sentence after the table was introduced.
Response: It has been corrected as far as possible. Figure 1 has been located just below the line 176. All tables have been positioned just below the corresponding paragraphs. All the corrections are highlighted in the manuscript by track changes.
6. Please introduce graphs to 3xplain tables.
Response: We have added two figures in supplementary material to connect all the parameters analyzed and avoid confusion. We firmly believe that the incorporation of these two figures (S1 and S2) provides improved robustness to this scoping review.

Reviewer 3 Report
It is very good that the authors were able to carry out a serious revision of the article. All my comments have been taken into account. The quality of the article has been significantly improved. The article can be accepted for publication in its current form.
Author Response
Reviewer #3,
It is very good that the authors were able to carry out a serious revision of the article. All my comments have been taken into account. The quality of the article has been significantly improved. The article can be accepted for publication in its current form.
Response: We would like to thank the reviewer for taking the necessary time and effort to revise this manuscript. We sincerely appreciate your valuable feedback, which helped us in improving the quality of the manuscript.

Round 3
Reviewer 1 Report
Thanks authors efforts. The manuscript has been improved. The figure S1 and S2 are excellent results.
Reviewer 1 asked great questions. It is hard to find all of responds through the manuscript. Please state your answers in figures.
- Which microbial were the most studied?
Please use the pie chart to illustrate what the percentage of each microbial in your study.
- How many types of SENPs were the most studied?
The column 3 in table 1 and 2 state Biogenic, Synthetic or Biogenic and Synthetic. Please move this column to column 1. Please use the pie chart to illustrate the what the percentage among all of your study.
- Which size of SeNPs is the most effect upon pathogenic oral microorganisms?
- What concentration of SeNPs range is the most effect?
You have made efforts and figure S1 can be employed to this answer this question. The table 1 is comprehensive with particle size and concentration tested. The figure S1 further illustrate of table 1. Please move figure S1 after the table 1. Line 236-line 250 are excellent results for the content of figure S1 and S2.
Table 1,2
Please move the first column reference to the last column in the table.
Author Response
Reviewer #1,
1. Thanks authors efforts. The manuscript has been improved. The figure S1 and S2 are excellent results.
Response: Thank you for your valuable feedback.
2. Reviewer 1 asked great questions. It is hard to find all of responds through the manuscript. Please state your answers in figures.
- Which microbial were the most studied?
Please use the pie chart to illustrate what the percentage of each microbial in your study.
Response: Thank you for this comment. As per your recommendation, we have added a pie chart showing the percentage of each oral microorganism studied in this scoping review. Please, see Figure S2a.
- How many types of SENPs were the most studied?
The column 3 in table 1 and 2 state Biogenic, Synthetic or Biogenic and Synthetic. Please move this column to column 1. Please use the pie chart to illustrate the what the percentage among all of your study.
Response: Thank you for your suggestion. We firmly believe that the changes made in tables improve their comprehension (please see the corrections in Tables 1,2,3 & 4). Moreover, as per your recommendation, we have added a pie chart showing the percentage of each type of SeNPs studied in this scoping review. Please, see Figure S1a.
- Which size of SeNPs is the most effect upon pathogenic oral microorganisms?
- What concentration of SeNPs range is the most effect?
- You have made efforts and figure S1 can be employed to this answer this question. The table 1 is comprehensive with particle size and concentration tested. The figure S1 further illustrate of table 1. Please move figure S1 after the table 1. Line 236-line 250 are excellent results for the content of figure S1 and S2.
Response: Thank you for this comment. However, we consider a little redundant to move Figure S1 to the main text. We prefer to keep this information on supplementary material section to be consulted whenever necessary.
3. Table 1,2
- Please move the first column reference to the last column in the table.
Response: We have made your recommended changes. The corrections are highlighted in the manuscript by track changes.
